# Rescue of Methionine Dependence by Cobalamin in a Human Colorectal Cancer Cell Line

**DOI:** 10.3390/nu16070997

**Published:** 2024-03-28

**Authors:** Sarita Garg, Isabelle R. Miousse

**Affiliations:** Department of Biochemistry and Molecular Biology, University of Arkansas for Medical Sciences, Little Rock, AR 72205, USA; gargsarita@uams.edu

**Keywords:** cobalamin, methionine, colorectal cancer, cell cycle

## Abstract

Methionine dependence is a characteristic of most cancer cells where they are unable to proliferate when the essential amino acid methionine is replaced with its precursor homocysteine in the growing media. Normal cells, on the other hand, thrive under these conditions and are referred to as methionine-independent. The reaction that adds a methyl group from 5-methyltetrahydrofolate to homocysteine to regenerate methionine is catalyzed by the enzyme methionine synthase with the cofactor cobalamin (vitamin B_12_). However, decades of research have shown that methionine dependence in cancer is not due to a defect in the activity of methionine synthase. Cobalamin metabolism has been tied to the dependent phenotype in rare cell lines. We have identified a human colorectal cancer cell line in which the cells regain the ability to proliferation in methionine-free, L-homocystine-supplemented media when cyanocobalamin is supplemented at a level of 1 µg/mL. In human SW48 cells, methionine replacement with L-homocystine does not induce any measurable increase in apoptosis or reactive oxygen species production in this cell line. Rather, proliferation is halted, then restored in the presence of cyanocobalamin. Our data show that supplementation with cyanocobalamin prevents the activation of the integrated stress response (ISR) in methionine-deprived media in this cell line. The ISR-associated cell cycle arrest, characteristic of methionine-dependence in cancer, is also prevented, leading to the continuation of proliferation in methionine-deprived SW48 cells with cobalamin. Our results highlight differences between cancer cell lines in the response to cobalamin supplementation in the context of methionine dependence.

## 1. Introduction

Both mammalian and non-mammalian cells have extensive systems to sense nutrient levels and adapt growth and metabolism accordingly. Those systems are frequently overactivated in cancer [1], leading to differential responses to nutrient deficiencies in cancer cells compared to normal cells. Methionine dependence is a characteristic of most cancer cells, where they are unable to proliferate when provided with homocysteine to replace the essential amino acid methionine. Normal cells, on the other hand, thrive under these conditions and are referred to as methionine-independent. In vivo, this translates into slower growth rates and decreased metastasis for tumors depleted in methionine through dietary [2,3,4] or enzymatic [5,6,7] means. Both normal and cancer cells respond to decreases in methionine availability by upregulating methionine adenosyltransferase MAT2A [8,9,10]. However, only methionine-dependent cells display a decrease in proliferation, an increase in oxidative stress, and apoptosis.

In addition to its role as a building block for protein synthesis, methionine is a precursor for methylation reactions. Methionine undergoes conversion into S-adenosylmethionine (SAM) catalyzed by MAT2A. SAM serves as the methyl donor for the methylation of DNA, histones, lipids, and other essential cellular molecules. Cancer is characterized by alterations to methylation patterns, including global hypomethylation, with hypermethylation of tumor suppressor genes. Similarly, repressive histone methylation marks are decreased globally and increased in tumor suppressor genes. Methionine is also a precursor for the antioxidant glutathione through the transulfuration pathway (reviewed in [11,12]). Consequently, methionine and its associated metabolites play a crucial role in carcinogenesis and cancer progression.

Early studies established that methionine dependence was not due to a deficiency in the enzyme methionine synthase, which remethylates homocysteine to methionine using the cofactor cobalamin (vitamin B_12_) [13,14]. Nevertheless, the metabolism of the cofactor cobalamin has been implicated in the methionine dependence phenotype. Methionine dependence has been associated with an increase in the expression of the transcobalamin receptor, which tethers the circulating transcobalamin–cobalamin complex to the plasma membrane and leads to its internalization [15]. Hypermethylation of the cobalamin chaperone gene *MMACHC* was also shown to induce methionine dependence in the melanoma cell line MeWo-LC1 compared to its methionine-independent parental cell line MeWo [16]. However, supplementation in cobalamin is not typically associated with any alleviation of the methionine dependence phenotype. In this project, we tested methionine-dependent colorectal cancer cell lines for their response to cobalamin. We identify and describe a human cell line where methionine independence is restored upon supplementation with cyanocobalamin and explore how it affects cellular endpoints and metabolism. Cobalamin did not significantly rescue proliferation in three other cell lines, highlighting individual differences in the role of cobalamin in the methionine dependence phenotype.

## 2. Materials and Methods

### 2.1. Cell Culture

SW48, HCT116, CT26, and MC38 cells were purchased from ATCC (Manassas, VA, USA). Cells were routinely passaged in standard DMEM media supplemented with 10% FBS and 100 IU penicillin and streptomycin PenStrep (Corning, Corning, NY, USA). For experiments, cells were cultured in high-glucose DMEM, no glutamine, no methionine, no cystine (ThermoFisher Scientific, Waltham, MA, USA) and supplemented with 10% dialyzed serum (BioTechne, Minneapolis, MN, USA), 100 IU penicillin, and streptomycin (ThermoFisher), 4 mM L-glutamine (ThermoFisher), and 1 mM sodium pyruvate (ThermoFisher). L-cystine (Millipore-Sigma, Burlington, MA, USA) was resuspended in PBS with NaOH added until complete solubilization and added to the cell media at a final concentration of 150 μM. For control medium, L-methionine (Millipore-Sigma) was resuspended in PBS and added to the cell media at a final concentration of 200 μM. For methionine-free homocysteine-containing media, L-homocystine (Chem-impex, Wood Dale, IL, USA) was resuspended in 1M HCl in PBS and added to the cell media at a final concentration of 200 μM. Homocystine is reduced to homocysteine in cells. Cyanocobalamin (Cbl) was added to the medium at a final concentration of 1 μg/mL (750 nM). The four treatment groups were as follows: CTL: 200 µM methionine, 0 µM homocystine, 0 nM Cbl, HCY: 0 µM methionine, 200 µM homocystine, 0 nM Cbl, CTL + Cbl: 200 µM methionine, 0 µM homocystine, 750 nM Cbl, HCY + Cbl: 0 µM methionine, 200 µM homocystine, 750 nM Cbl.

### 2.2. Proliferation

We plated cells at a density of 300,000 per well in triplicates in 6-well plates in standard DMEM. After 3 h, we washed the cells once with PBS then added the test media. We incubated the cells 24 or 48 h. At the end of the incubation period, we collected the media and trypsinized cells, rinsed the cell pellet once in PBS, then performed a trypan blue live/dead count with an automated cell counter (Countess 3, ThermoFisher). 

### 2.3. Apoptosis

We plated, treated, and collected cells as before, with 3 extra wells for compensation (unstained, Annexin V only, and PI only). We resuspended the rinsed cell pellet in 150 µL annexin V staining solution (FITC-conjugated antibody diluted 1:100 in binding buffer (140 mM NaCl, 10 mM HEPES pH 7.4 and 2.5 mM CaCl_2_)) and incubated at room temperature for 20 min. We then added 100 µL propidium iodide (PI) staining buffer (20 ng/mL propidium iodide (ThermoFisher) in binding buffer). We measured fluorescence by flow cytometry on a BD LSRFortessa Cell Analyzer (BD Biosciences, Franklin Lakes, NJ, USA) and analyzed the results with FlowJo software version 10.9.0 (BD Biosciences).

### 2.4. Gene Expression

For real-time quantitative PCR assessment of gene expression, we plated and treated the cells as before and lysed the cells in Qiazol buffer (Qiagen, Germantown, MD, USA). We extracted RNA according to the manufacturer’s instructions, adding a second 70% ethanol wash. RNA quality and quantity was assessed by spectrophotometry (Nanodrop, ThermoFisher). Reverse transcription was performed on 1 μg of purified RNA using the iScript Reverse Transcription Supermix (Bio-Rad, Hercules, CA, USA). For each quantitative real-time PCR reaction, 20 ng of cDNA was used and amplified with the iTaq Universal SYBR Green Supermix (Bio-Rad).

### 2.5. Protein Expression

For Western blot analysis of protein expression, we plated, treated, and collected cells as before. We lysed the cells in RIPA buffer with phosphatase and protease inhibitors. We prepared the lysates in Laemmli buffer and loaded 20 µg of protein per lane on a Tris-Glycine gel. After running, we transferred the gel unto a PVDF membrane, blocked with 1% casein, and incubated in primary antibody (1:1000) overnight: MAT2A ab177484, MMUT ab134956 (Abcam, Waltham, MA, USA) and MTR 25896-1-AP (Proteintech, Rosemont, IL, USA). We then used a goat anti-rabbit IgG secondary antibody for fluorescent detection (#12004162, Bio-Rad), along with a rhodamine-conjugated anti-actin primary antibody as a loading control (#12004164, Bio-Rad). We acquired the images on a ChemiDoc Imaging system and performed the analysis with Image Lab version 6.1.0 (Bio-Rad).

### 2.6. Cell Cycle

We plated cells at a density of 300,000 per well in triplicates in two 6-well plates in standard DMEM plus one well for the unstained controls. After 3 h, we washed the cells once with PBS then added the test media. After 48 h, we harvested the cells with trypsin, rinsed them in PBS, then fixed the cells in 70% ethanol overnight. On the day of the analysis, we washed the cells twice in PBS, then resuspended the cell pellets in 50 µL of a 100 µg/mL RNase A solution. We then added 200 µL of a 1 µg/mL DAPI solution (except the unstained control; 200 µL PBS). We measured fluorescence by flow cytometry on a BD LSRFortessa Cell Analyzer (BD Biosciences, Franklin Lakes, NJ, USA) and analyzed the results with FlowJo software version 10.9.0 (BD Biosciences).

### 2.7. Data Analysis

The software GraphPad Prism 10.1.2 (GraphPad Software, San Diego, CA, USA) was used to represent the data graphically and perform statistical analysis. For the RNA expression, we show the mean ± standard deviation of four replicates for each condition in each cell line. Comparisons were made using a two-way ANOVA with Šídák’s multiple comparison test. Figures were created with Inkscape 1.2 (Inkscape Project).

## 3. Results

### 3.1. Cobalamin Supplementation Rescues Cell Number in Methionine-Starved SW48 Human Colorectal Cancer Cells

Due to the role of cobalamin in the metabolism of methionine, we analyze the effect of cobalamin supplementation on methionine dependence in three colorectal cancer cell lines. The human HCT116 and the murine MC38 and CT26 showed a strong methionine dependence phenotype as assessed by the live cell count when we replaced methionine with L-homocystine in the culture media for 48 h (Figure 1). Cyanocobalamin supplemented at 1 µg/mL (750 nM) did not increase the cell number in homocystine-treated HCT116 and MC38 cells (Figure 1A,B), and a small near-significant increase in CT26 cells was observed (Figure 1C). The human SW48 cell line displayed a milder methionine-dependent phenotype, with a significant decrease in the cell count in homocystine-treated versus methionine-treated cells, but this decrease was less pronounced than in HCT116, MC38, and CT26. Interestingly, addition of cobalamin restored the cell number to the same level seen in the methionine-treated cells (Figure 1D). Functionally, cobalamin supplementation was able to convert methionine-dependent SW48 cells to methionine independence. However, this was not seen in three additional colorectal cancer cell lines, indicating that this mechanism of rescue is specific to SW48.

### 3.2. Apoptosis Is Not Induced by Homocystine Treatment in SW48 Cells

The cell numbers measured above are the results of the effect of proliferation and cell death. We and others have previously reported that cell death in methionine dependence is linked to apoptosis [17,18,19]. The homocystine treatment was associated with a small increase in apoptosis at 48 h, which did not reach significance (Figure 1E,F). However, the combination of homocystine treatment with cobalamin induced a small but significant increase in apoptosis compared to controls. This was seen both in early apoptosis (annexin V positive, PI negative cells, Figure 1E), and late apoptosis (Annexin V positive, PI positive cells; Figure 1F). We also measured apoptosis in the HCT116 cell line. The addition of cobalamin did not significantly change the proportion of apoptotic cells in that cell line in either control nor homocystine media (Appendix A). These results suggest that the cobalamin-dependent rescue observed in SW48 cells is not due to reduced apoptosis.

### 3.3. Cobalamin Does Not Reduce Oxidative Stress

We hypothesized that cobalamin may reduce oxidative stress to rescue proliferation. We indeed observed that the expression of the oxidative stress-related gene Heme Oxygenase 1 (HMOX1) was increased in homocystine compared to methionine and that addition of cobalamin restored the expression at control levels (Figure 1G). Cells grown in homocystine displayed a slightly higher signal level for the oxidative stress marker CellROX than cells grown in methionine, but this did not reach significance (Figure 1H). This increase was not rescued by cobalamin. On the opposite, cobalamin further increased oxidative stress. We were able to confirm that there was no difference in the ratio of mitochondrial stress (MitoRed) to mitochondrial mass (MitoGreen) in cells grown in homocystine versus methionine (Appendix A). We conclude that cobalamin rescues proliferation in methionine-starved SW48 cells by a mechanism other than reduction of oxidative stress. 

### 3.4. Cobalamin Prevents the Activation of the Integrated Stress Response in Homocystine-Treated SW48 Cells 

The gene HMOX1 (Figure 1G), in addition to being linked to oxidative stress, is also known to be upregulated during the integrated stress response (ISR). We and others have demonstrated that the ISR is activated during methionine starvation [17,20,21,22]. We therefore analyzed further genes associated with the ISR and the transcription factor Activating Transcription Factor 4 (ATF4). As expected, ATF4 gene expression was upregulated in the homocystine group compared to the methionine control group (Figure 2A). With the addition of cobalamin, there was still a significant increase in the homocystine group compared to methionine, but only of 1.2 folds compared to 2.8 folds in the comparison without cobalamin. The ATF4-regulated transcription factor DNA Damage-Inducible Transcript 3 (DDIT3/CHOP) similarly showed a 1.9-fold increase in the homocystine group compared to the methionine control, but no difference between the two cobalamin-treated groups (Figure 2B). The same pattern was found in other ATF4-associated transcripts such as the stress-associated growth factor Fibroblast Growth Factor 21 (FGF21) (Figure 2C), and the methionine transporter Solute Carrier Transporter 7A5 (SLC7A5) (Figure 2D). Interestingly, it was also the case for the lysosomal cobalamin exporter ATP Binding Cassette subfamily D member 4 (ABCD4) [23], which has not traditionally been associated with ATF4 (Figure 2E). In all cases, the calculated *p* value for the effect of homocystine, cobalamin, and their interaction was less than 0.0001. These data indicate that cobalamin supplementation prevents the activation of genes downstream of the ISR in response to the substitution of methionine with homocystine in SW48 cells. 

### 3.5. Cobalamin Rescues Cell Cycle Arrest in Methionine-Starved SW48 Cells

Under moderate ER stress, ATF4 activation induces an adaptive cell cycle arrest [17,24,25]. We hypothesized that by preventing the induction of the ISR, cobalamin would relieve the cell cycle arrest. Indeed, the data show an increase in the proportion of cells in G1 when methionine is replaced with homocystine, and this increase is counteracted by cobalamin (Figure 2F). In parallel, the proportion of cells in S phase plunges with the homocystine treatment compared to methionine, but not in the homocystine with cobalamin group (Figure 2G). These data indicate that cobalamin allows SW48 cells to overcome the G1 arrest in homocystine.

### 3.6. Effects on Protein Expression

The Catalogue of Somatic Mutations in Cancer (COSMIC) reports that SW48 is a heterozygote for the G426R thermolabile methylmalonyl-CoA mutase (MMUT) mutation [26] (https://cancer.sanger.ac.uk/cosmic/sample/overview?id=2302017#mut-spec, accessed on 25 February 2024). However, the presence of cobalamin did not increase MMUT protein expression in our samples (Appendix A). There was also no change in MMUT gene expression (Appendix A). On the other hand, we did detect a slight increase in the protein abundance of methionine synthase (MTR) by 33% in the cobalamin-supplemented samples (Appendix A). We did not detect any change in MTR at the gene expression level (Appendix A). The cobalamin-mediated increase in MTR levels did not, however, prevent the upregulation of the enzyme methionine adenosyltransferase 2A (MAT2A), a good marker of methionine deficiency [8], in the homocystine-treated groups (Appendix A). The pattern at the gene expression level was similar for MAT2A (Appendix A).

## 4. Discussion

Methionine dependence has been reported in a wide range of cancer types. Despite the role of methionine synthase and cobalamin in the conversion between homocysteine and methionine, the literature over the past 40 years does not support a major role for a deficiency in methionine synthase function in methionine dependence in cancer. Nevertheless, recent data show the complete rescue of proliferation in methionine-starved liver cancer cells when cobalamin was added [27]. We investigated the effect of cobalamin supplementation on methionine dependence in a colorectal cancer model.

We contrasted proliferation in four cell lines; two human and two murine. Two of the four cell lines showed no indication of rescue by cobalamin (HCT116 and MC38), one cell line showed a near significant rescue (CT26), and one single cell line showed rescue back to control levels; SW48. Our data indicate that complete rescue of proliferation by cobalamin is not a widespread phenomenon and may vary by cancer type.

In SW48 cells, methionine dependence does not induce significant apoptosis nor oxidative stress. We conclude that the rescue by cobalamin does not occur through these two mechanisms. However, we did observe a change in gene expression pattern consistent with the ISR in methionine-starved SW48 cells. This ISR signature was completely eliminated with cobalamin supplementation. Interestingly, the literature does not support a role for the kinase GCN2 and the phosphorylation of its target eIF2α in the activation of ATF4 and the ISR in methionine-deprived cells [20].

Activation of the ISR under moderate ER stress contributes to cell survival by inducing an arrest of the cell cycle [24,25]. In methionine dependence in particular, G1 arrest has been well described in previous work [22,28]. This has been attributed to the destabilization of pre-replication complexes [29]. In this project, we observed that the rescue of the methionine dependence phenotype with cobalamin was associated with a return of the cell cycle back to control values. Specifically, we saw a normalization of the percentage of cells in the S phase and G1 phase with cobalamin supplementation.

There are two cobalamin-dependent enzymes in mammals—MTR and MMUT. The presence of cobalamin did not promote the protein abundance of MMUT, but there was a slight increase in MTR in both cobalamin-treated groups by about one third. This was not seen at the gene expression level, indicating that cobalamin may increase the stability of the protein as suggested previously [30]. An increase in abundance in MTR could potentially promote conversion of homocysteine to methionine and rescue the phenotype. Under these circumstances, we would expect methionine levels to increase. Methionine levels are tightly associated with MAT2A expression in cells [8,31]. However, MAT2A remained overexpressed in homocysteine-treated samples supplemented with cobalamin, suggesting that methionine is still depleted in these cells.

Our results highlight that altered cell cycle is a fundamental component of methionine dependence in cancer, as opposed to increased apoptosis and oxidative stress. The data also raise many questions about the frequency and characteristics of cobalamin-responsive cancer cell lines. The literature suggests that the phenomenon may be more frequent in liver cancer cells [27]. This hints at a possible involvement of the transulfuration pathway, which is active in hepatic tissues. The limited scope of our study, with only four cell lines in a single cancer type, suggests that an expanded screen of cobalamin responsiveness may reveal further insights into one-carbon metabolism.

## 5. Conclusions

Our study investigated the ability of cobalamin, a vitamin crucial for methionine synthesis, to rescue cancer cells from methionine dependence. One out of four tested colorectal cancer cell lines was rescued by supplementation with cobalamin. The rescue was independent of both apoptosis and oxidative stress, but was related to the alleviation of G1 arrest. These data suggest that cobalamin may be able to rescue a mild methionine dependence phenotype by promoting DNA synthesis and the maintenance of the cell cycle.

## Figures and Tables

**Figure 1 nutrients-16-00997-f001:**
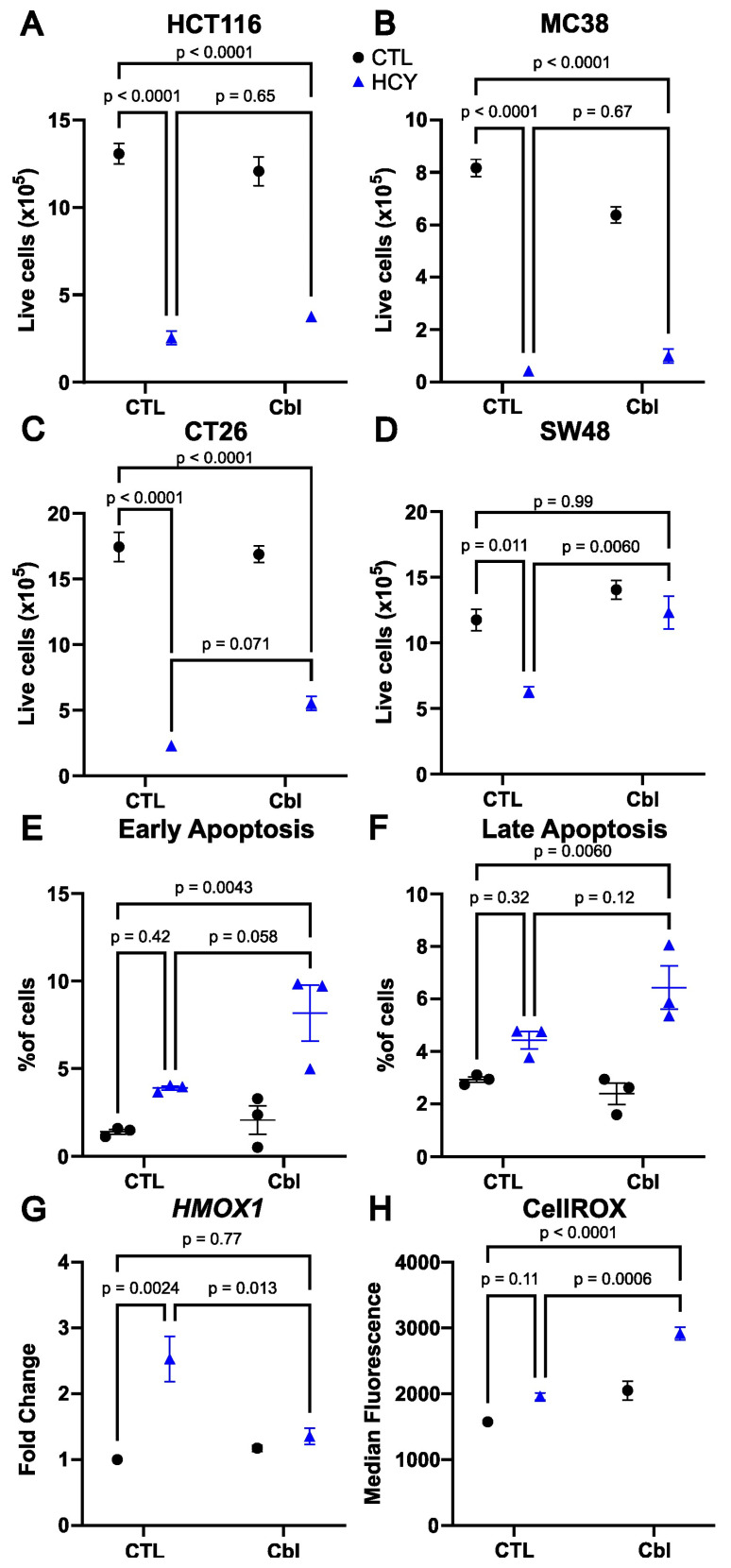
Cobalamin rescues SW48 cells without changes in apoptosis or oxidative stress. Live cell number at 48 h in HCT116 (**A**), MC38 (**B**), CT26 (**C**), and SW48 (**D**) colorectal cancer cells. Percentage of cells undergoing early (annexin V positive, PI negative; (**E**), and late apoptosis (Annexin V positive, PI positive; (**F**) and *HMOX1* gene expression (**G**) and CellROX oxidative stress reporter assay (**H**) in SW48 cells. CTL: Control, Cbl: cobalamin, HCY: homocystine (Blue). Two-way ANOVA with Šídák’s multiple comparison test; standard error of mean and three of the six comparisons shown.

**Figure 2 nutrients-16-00997-f002:**
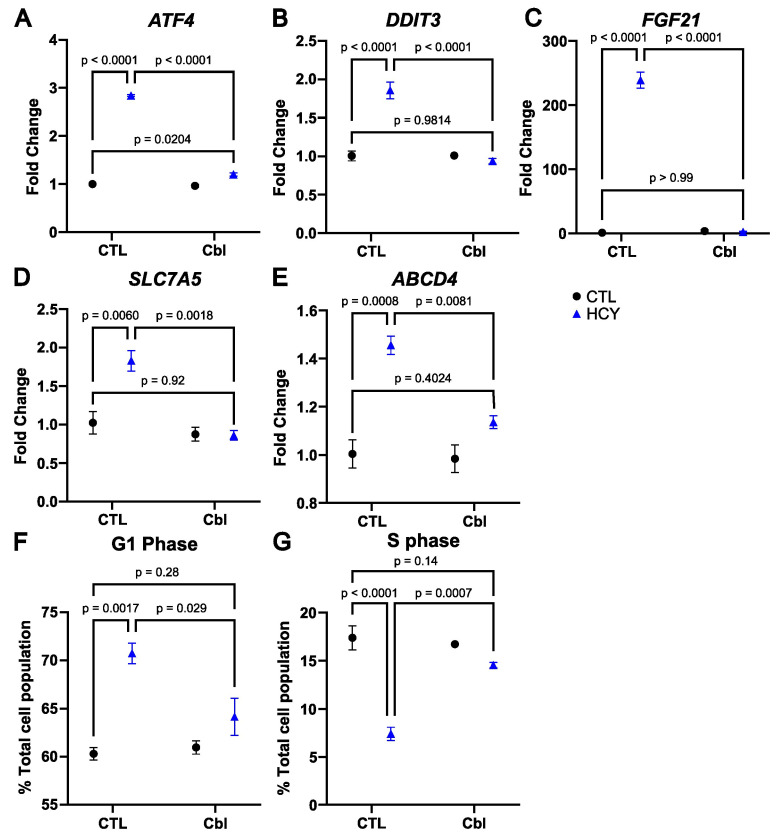
Cobalamin prevents the activation of the integrated stress response and restores the cell cycle in SW48 cells. Expression of the genes *ATF4* (**A**), *DDIT3* (**B**), *FGF21* (**C**), *SLC7A5* (**D**), and *ABCD4* (**E**). Percentage of the total cell population in G1 phase (**F**) and S phase (**G**). CTL: Control, Cbl: cobalamin, HCY: homocystine (Blue). Two-way ANOVA with Šídák’s multiple comparison test; standard error of mean and three of the six comparisons shown.

## Data Availability

The original contributions presented in the study are included in the article/Appendix A, further inquiries can be directed to the corresponding authors.

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
