# Peer review of "Rescue of Methionine Dependence by Cobalamin in a Human Colorectal Cancer Cell Line"

_nutrients, 2024, doi:10.3390/nu16070997_

Round 1

Reviewer 1 Report

Comments and Suggestions for Authors

This paper describes that cobalamin restores methionine dependence in SW48 cells, in which ER stress may be involved, as methionine deficiency (replacing with homocysteine) induced several IRS genes and caused cell cycle arrest. Although the molecular mechanism remains unclear, the finding may be important for understanding cancer cell metabolism. The experiments have been done well, and the conclusions are almost supported by experimental data convincingly.

1) In Fig. 2: To confirm whether the treatment of cobalamin in methionine-starved condition cause apoptosis in other cell lines that failed to be rescued by cobalamin (as shown in Fig. 1), similar experiments should be performed using cell lines such as HCT116 and MC38. This may rule out the possibility that the increase in survival with cobalamin was offset or diminished by apoptosis.
2) Line 223: The description of "a very slight but significant rescue (CT26)" is inappropriate, because the line 138 indicates “"a small near significant increase in CT26" with the p-value of 0.071 (Fig.1C).

Minors:
3) Line 64: the information on CT26 is missing.
4) Line 95: Indicate the company for Qiazol buffer.
5) Figs 4-6: Indicate SW48 cells as a used cell line in these experiments.
6) Fig.6A: Label each lane.

Author Response

We would like to thank the reviewers for their time and effort in reviewing our manuscript. We have found the comments to be helpful and hope that we have matched the reviewers’ expectations. The changes have been highlighted in blue in the revised file.

Reviewer 1

This paper describes that cobalamin restores methionine dependence in SW48 cells, in which ER stress may be involved, as methionine deficiency (replacing with homocysteine) induced several IRS genes and caused cell cycle arrest. Although the molecular mechanism remains unclear, the finding may be important for understanding cancer cell metabolism. The experiments have been done well, and the conclusions are almost supported by experimental data convincingly.

1) In Fig. 2: To confirm whether the treatment of cobalamin in methionine-starved condition cause apoptosis in other cell lines that failed to be rescued by cobalamin (as shown in Fig. 1), similar experiments should be performed using cell lines such as HCT116 and MC38. This may rule out the possibility that the increase in survival with cobalamin was offset or diminished by apoptosis.

Response: We think this is a very valid point that the reviewer brought. This experiment was quite straightforward to do and we can report that cobalamin did not bring any change to the level of apoptosis in HCT116 with either control or homocystine media. We included the data in supplementary figure 1 with a comment in the text in section 3.2.

2) Line 223: The description of "a very slight but significant rescue (CT26)" is inappropriate, because the line 138 indicates “"a small near significant increase in CT26" with the p-value of 0.071 (Fig.1C).

Response: We apologize for the discrepancy and have made the appropriate modification.
Minors:
3) Line 64: the information on CT26 is missing.
4) Line 95: Indicate the company for Qiazol buffer.
5) Figs 4-6: Indicate SW48 cells as a used cell line in these experiments.
6) Fig.6A: Label each lane.

Response: We incorporated the changes suggested.      

Reviewer 2 Report

Comments and Suggestions for Authors

The aim of this study was to “describe a human colorectal cancer cell line where methionine independence is restored upon supplementation with cyanocobalamin and to explore how it affects cellular endpoints and metabolism”. This brief report is interesting and the results are promising. In the abstract of the paper, I recommend clearly writing the purpose of the paper, (as in the last sentence of the Introduction subsection). The introduction of the work is written logically, I have no substantive comments. The authors seem to have experience in this type of research, citing their prior work on the same topic. What is new about this work, please highlight more. The subsection "Material and methods", is written rather briefly, and not everything is clear. In 2.1. please clearly write what groups of cells studied were divided, it is not clear from the figures obtained. I would only add the names of the techniques used in the paper, e.g. flow cytometry - 2.3.; in subsection 2.4. please add that RT-PCR technique was used, and in 2.5. please add that protein expression was studied using Western blot technique.

The results are described sequentially and illustrated in the following Figures 1-2. I think it would be better for the overall paper, to place the Supplementary Figures 1-2 within the main text.

Please standardize the citation throughout the paper, either all in bold, or all citations without bolding and without italics (page 7, lines: 152, 188,194, 204, etc.].

I would still suggest checking that all abbreviations in the paper are explained when they first appear, e.g. PI (line 89), HMOX1 (line 161), ATF4 (line 179), DDIT3 (line 183), FGF21 and SLC7A5 (lines 186-187), etc.

The discussion, although brief, discusses the essential results of the work, the conclusions correspond to the results of the work. The only thing not written are the limits of the work, further plans, what could still be done in this field. The literature cited is about 40% from the last 5 years.

Author Response

We would like to thank the reviewers for their time and effort in reviewing our manuscript. We have found the comments to be helpful and hope that we have matched the reviewers’ expectations. The changes have been highlighted in blue in the revised file.

The aim of this study was to “describe a human colorectal cancer cell line where methionine independence is restored upon supplementation with cyanocobalamin and to explore how it affects cellular endpoints and metabolism”. This brief report is interesting and the results are promising. In the abstract of the paper, I recommend clearly writing the purpose of the paper, (as in the last sentence of the Introduction subsection). The introduction of the work is written logically, I have no substantive comments. The authors seem to have experience in this type of research, citing their prior work on the same topic. What is new about this work, please highlight more. The subsection "Material and methods", is written rather briefly, and not everything is clear. In 2.1. please clearly write what groups of cells studied were divided, it is not clear from the figures obtained. I would only add the names of the techniques used in the paper, e.g. flow cytometry - 2.3.; in subsection 2.4. please add that RT-PCR technique was used, and in 2.5. please add that protein expression was studied using Western blot technique.

Response: We would like to thank the reviewer for the suggestions. We expanded the abstract and introduction, described the biological groups, and named out the different techniques used.

The results are described sequentially and illustrated in the following Figures 1-2. I think it would be better for the overall paper, to place the Supplementary Figures 1-2 within the main text.

Response: The “Brief Report” format is limited to 2 figures, which explains why we presented the information as supplementary data. Given these limitations, we would be happy to discuss with the reviewer and assistant editor of possible arrangements for the figures. https: //www. mdpi. com/about/article_types

Please standardize the citation throughout the paper, either all in bold, or all citations without bolding and without italics (page 7, lines: 152, 188,194, 204, etc.].

Response: We appreciate the attention to details and made the suggested modifications.

I would still suggest checking that all abbreviations in the paper are explained when they first appear, e.g. PI (line 89), HMOX1 (line 161), ATF4 (line 179), DDIT3 (line 183), FGF21 and SLC7A5 (lines 186-187), etc.

Response: We made the suggested modifications

The discussion, although brief, discusses the essential results of the work, the conclusions correspond to the results of the work. The only thing not written are the limits of the work, further plans, what could still be done in this field. The literature cited is about 40% from the last 5 years.

Response: The reviewer will find an additional discussion paragraph addressing these topics.